# Investigation of Spontaneous Combustion Zones and Index Gas Prediction System in Goaf of "Isolated Island" Working Face

**Jiamei Chai** 

**Abstract:** Studies show that accurate division of spontaneous combustion (SC) zones in the goaf and the determination of the prediction system of the SC index are of great significance to prevent spontaneous and unwanted combustions in the goaf. Aiming at resolving the problem of coal SC in the goaf of an "isolated-island" fully mechanized caving face, a multiphysics model coupled with gas flow field and gas concentration field was established in the present study. Taking the 8824 working face of Nanzhuang coal mine as the research object and the oxygen concentration as the division index, coal SC was simulated in the goaf. The obtained results show that the ranges of heat dissipation zone, oxidation zone, and the asphyxia zone on the air inlet side are around 0–107 m, 107–239 m, and beyond 239 m, respectively. Moreover, the ranges of the three zones on the return air side are 0–13 m, 13–189 m, and beyond 189 m, respectively. The ranges of the three zones in the middle of goaf are 0–52 m, 52–213 m, and beyond 213 m, respectively. The performed analyses demonstrate that the obtained simulation results are consistent with the experimental data. Meanwhile, the coal programmed temperature rise experiment was carried out to improve the prediction index gas system of SC. It was found that CO and $C_2H_4$ can be used as early warning indices of SC in the goaf, while $C_2H_6$, $C_3H_8$, and $C_2H_4/C_2H_6$ are auxiliary indices to master the coal SC.

**Keywords:** isolated island working face; numerical simulation; "three zones" of spontaneous combustion in goaf; beam tube monitoring; index gas



## 1. Introduction

Coal mine fire is a catastrophic phenomenon that occasionally occurs and affects the safety of workers and reduces the production efficiency. Studies show that coal mine fires mainly initiate from spontaneous combustion (SC) in a goaf that spreads rapidly and may even result in serious thermal dynamic disasters [1]. Generally, coal mine fire is hidden, sudden, difficult-to-control, and easily causes secondary disasters such as gas and coal dust explosions [2–6]. Further investigations reveal that this catastrophe not only produces huge amounts of toxic gases that endanger the safety of workers but also affect the mining efficiency [7]. Recently, problems such as excessive dangerous gas in the working face and goaf fire caused by coal residue have concerned many scholars. These concerns are especially more pronounced in the "isolated-island" working face of a fully mechanized coal mine. The goaf of the isolated-island working face is surrounded by other goafs that leads to the air leakage channel of the goaf of the isolated-island working face, which is more complex and changeable than that of the conventional goaf, and the ground pressure is strong. The roadway has large compression deformation, stress concentration, and frequent roof falling accidents, resulting in slow advancing speed of the working face, and the coal of the working face is broken, resulting in more loose coal being left in the goaf, which increases the risk of oxidation, heat storage, and SC of residual coal in the goaf, thereby initiating a catastrophe [8,9]. The problem of CO concentration exceeding the limit is becoming more and more prominent, even far exceeding the provisions of "Safety Regulations in Coal Mines of China", which will seriously threaten the life safety of

front-line coal mine workers [10]. Accordingly, it is of significant importance to investigate SC in goaf originating from residual coal.

Domestic and foreign scholars have carried out extensive and in-depth research on the problem of spontaneous combustion in the goaf. Among them, in terms of the division of three zones of spontaneous combustion in the goaf, Deng et al. [11,12] obtained three-dimensional temperature profiles in the goaf and divided the goaf into three SC zones. Finally, the PSO–SVR model was proposed to calculate the seam temperature in the coal SC. Li et al. [13–15] simulated the goaf by means of numerical simulation, analyzed the influence of the oxygen concentration, heat storage area, and various boundary conditions on the SC zones in the goaf. Moreover, Hao et al. [16] performed experiments and investigated the influence of ventilation and air leakage on the oxygen concentration in the goaf. Accordingly, high-risk areas were determined from the aspect of coal SC. The above scholars have divided the three zones of SC in the goaf by means of numerical simulation and the establishment of an experimental platform, using the oxygen concentration as an indicator, but they have not verified the field and cannot accurately divide the three zones of SC in the goaf.

On the aspect of early warning of the signs of SC gas in the goaf, Gao et al. [17,18] analyzed the CO concentration in the mining face and studied the working face in the oxidation process. Lin et al. [19] considered the coal volume as a porous system and modified the standard model by combining the conservation of species and energy equation and simulated the system under steady-state conditions. The starting, ending, and point temperatures were determined based on the expansion mechanism of the coal combustion. It was found that as the air inlet speed increases, the corresponding position of the critical point moves inward. Qin [20] took the fire coefficient as an auxiliary index parameter, fitted the distribution of the marker gas concentration against temperature, and proposed a method to calculate the ambient temperature of residual coal. Zhu et al. [21–23] comprehensively analyzed the detection temperatures of different index gases, including CO, $C_2H_4$, and $C_2H_6$, and detected the situation of the fire area using the CO index method. The above scholars believe that CO concentration can be used as a key indicator for predicting SC. However, single indicator gas monitoring is not sufficient for complex mine fire monitoring, and mine airflow will interfere with gas concentration and affect prediction accuracy. Therefore, other index parameters are urgently needed to assist coal SC prediction in the goaf.

In addition, the performed investigations demonstrate that air leakage is the main problem of SC in the goaf. In other words, accurate control of the air inlet volume and early warning are the key technical problems around preventing SC in the goaf. In this regard, the determination method of the marker gas was applied to monitor the air leakage in the goaf of a fully mechanized top coal caving face, and the correlation between air leakage and SC was obtained [24,25]. Moreover, the distribution of gas components and landmark gases was analyzed in the goaf and a correlation was established between temperature, oxygen content, and landmark gases [26–28]. Accordingly, in order to effectively prevent goaf SC, the stage and trend of coal SC can be determined by monitoring the index gas, so as to further optimize the prediction and prevention of coal combustion, and manage the risk of coal SC through the change of gas index.

In the present study, the multiphysical field coupling model is established to study coal SC in the goaf of the "isolated island" working face. The model is then applied to analyze the air inlet velocity in goaf, and the scope of "three zones" of spontaneous combustion in the goaf is divided with oxygen concentration as the index, validated by on-site bundled tube monitoring tests. Meanwhile, the symbolic gas in the goaf is measured and analyzed to determine the index gas early warning system of the coal SC, which is of great significance to the improvement of the control system of coal SC in the goaf of the isolated-island working face.

## 2. Project Overview

Nanzhuang coal mine is located in the south of the suburb of Yangquan City, Shanxi Province, China. The main mining coal seam number 15 is 5.24–6.63 m thick, with an average thickness of 5.89 m. It contains approximately 1–2 layers of gangue. The thickness of the gangue layer is about 0.02–0.47 m. The roof and floor of the coal seam are mudstone and sandy mudstone, respectively. Moreover, the working face 8824 is 183 m long with a 1394-m advancing strike and is buried at a depth of 500 m. The fully mechanized top coal caving method was used for mining. Since the mining of working faces 8822 and 8826 on the left and right sides of the working face, and the mining of working faces 4606 and 4607 corresponding to the upper number 12 coal seam has been completed, the working face 8824 has become a typical isolated-island fully mechanized top coal caving face. Moreover, Nanzhuang coal mine is a high-gas mine, and the mining of the isolated-island face brings great challenges to the prevention and control of gas and natural ignition.

## 3. Mathematical Model

### 3.1. Theory of the Goaf Flow Field

The conservation of momentum for an incompressible flow between the working face and the goaf can be expressed using the following Navier–Stokes equations:

$$\rho(u{\cdot}\nabla)u = \nabla{\cdot}[-pI + (\mu + \mu_T)(\nabla u + (\nabla u)^T] + F \tag{1}$$

The dynamic viscosity is defined as follows:

$$mu = \frac{\mu}{\rho} \tag{2}$$

When the airflow in the goaf is excessive airflow, the governing equations can be expressed in the form of a Brinkman equation:

$$(\mu/k)u = \nabla{\cdot}[-pI + (1/\varepsilon_P)\mu(\nabla u + (\nabla u)^T] + F \tag{3}$$

where $\rho$ is the air flow density, kg/m$^3$, $\mu$ is the dynamic viscosity of the air flow, Pa·s, $I$ is the identity matrix, $p$ is the air flow pressure, Pa, $T$ is the tunnel air flow temperature, °C, $\varepsilon_P$ is the porosity, 1, $k$ is permeability, m$^2$, $F = 12.25\sin(\alpha){\cdot}y{\cdot}(T - T_0)/T$ denotes the volumetric force originating from the inclination of the working face and physical properties of the air flow, including temperature, humidity, and density, N/m$^3$, $\alpha$ is the average dip angle, °, and $y$ is the width of fully mechanized top coal caving face, m.

### 3.2. Gas Concentration in Goaf

Considering the convection–diffusion of gas in the goaf, the governing equations can be expressed as follows [10]:

$$\frac{\partial c}{\partial t} + \nabla{\cdot}(-D\nabla c) + \boldsymbol{u}{\cdot}\nabla c = R \tag{4}$$

where $c$ is the gas concentration, mol/m$^3$, $\boldsymbol{u}$ is the vector flow rate of gas flow, m/s, $D$ is the diffusion coefficient, m$^2$/s, and $R$ denotes the gas reaction rate, mol/(m$^3$·s).

### 3.3. Determination of Key Parameters of the Numerical Simulation

(1) Coefficient of dilatancy

Based on the overlying rock movement theory [29], the goaf can be divided into three areas, including the natural accumulation area, the load-affected area, and the compaction stability area.

In the natural accumulation area, the crushing expansion coefficient $K_{P_1}$ is defined as follows:

$$K_{P_1} = \frac{\sum h - \Delta h + m_1 + m_2[1 - (1 - C)K_{Pc}]}{\sum h} \tag{5}$$

Moreover, the rock mass expansion coefficient $K_{P_2}$ in the load-affected zone can be expressed as follows:

$$K_{P_2} = \frac{\sum h - \Delta h + m_1 + m_2}{\sum h} - \frac{L_0(1 - C)}{\sum h(L_0 - L)} m_2 K_{Pc} \tag{6}$$

The rock mass expansion coefficient $K_{P_3}$ in the compaction stability zone can be expressed as follows:

$$K_{P_3} = \frac{\sum h - \Delta h + m_1 + m_2}{\sum h} - \frac{L_0(1 - C)}{\sum h(L_0 - L)} m_2 \tag{7}$$

where $m_1$ and $m_2$ denote the caving height and the coal drawing of fully mechanized top coal, respectively. Furthermore, $\sum h$ is the direct top thickness and $\Delta h$ reflects the gap between the rock mass and the roof in the collapse zone. $C$ is the recovery rate of top coal, and $K_{P_i}$ is the coefficient of crushed swelling of residual coal in the goaf, $L$ is the top control distance of the working face, and $L_0$ is the periodic pressure step.

(2) Porosity

Studies show that the porous distribution in the goaf strongly depends on the coefficient of dilatancy. More specifically, the greater the coefficient of dilatancy, the greater the porosity. This can be mathematically expressed as follows:

$$\varepsilon_P = 1 - \frac{1}{K_p} \tag{8}$$

(3) Permeability

The permeability of the goaf can be determined using the distribution of porous fraction and the Kozeny–Carman (KC) expression [30].

$$k = \frac{\varepsilon_p^3 \overline{d}^2}{150(1 - \varepsilon_p)^2} \tag{9}$$

(4) Oxygen diffusion coefficient

Under different temperatures and pressures, the oxygen diffusion coefficient in loose coal can be obtained from the expression below:

$$D = (0.8011\varepsilon_p - 0.1616)D_0 \left(\frac{T}{T_0}\right)^{\frac{2}{3}} \frac{P_0}{P} \tag{10}$$

(5) Oxygen consumption rate

Studies [12] reveal that the oxygen consumption rate in the coal combustion can be obtained from the following expression:

$$R_{O_2} = -\frac{c_{O_2}}{c_{O_2}^0} \gamma_0 e^{a(T - T_0)} \tag{11}$$

where $\overline{d}$ is the average particle size, mm, $D_0$ is the diffusion coefficient of air at a normal temperature and pressure, $m^2/s$, $R_{O_2}$ is the oxygen consumption rate, $mol/(m^3 \cdot s)$, $c_{O_2}^0$ and $c_{O_2}$ denote the initial and instantaneous oxygen concentration, $mol/m^3$, respectively. The oxygen consumption coefficient is represented by $\gamma_0$, $mol/(m^3 \cdot s)$; $T_0$ and $T$ reflect the

initial temperature of the coal seam and the coal temperature, K, respectively. Finally, *a* is the oxidation index of coal at the test conditions, $K^{-1}$.

## 4. Model Results and Analysis

### 4.1. Simulation Cases and Conditions

In the present study, the Nanzhuang mine is simulated as the research object and the influence of wind speed on the three zones of the goaf is analyzed. Figure 1 shows that the computational domain consists of four parts.

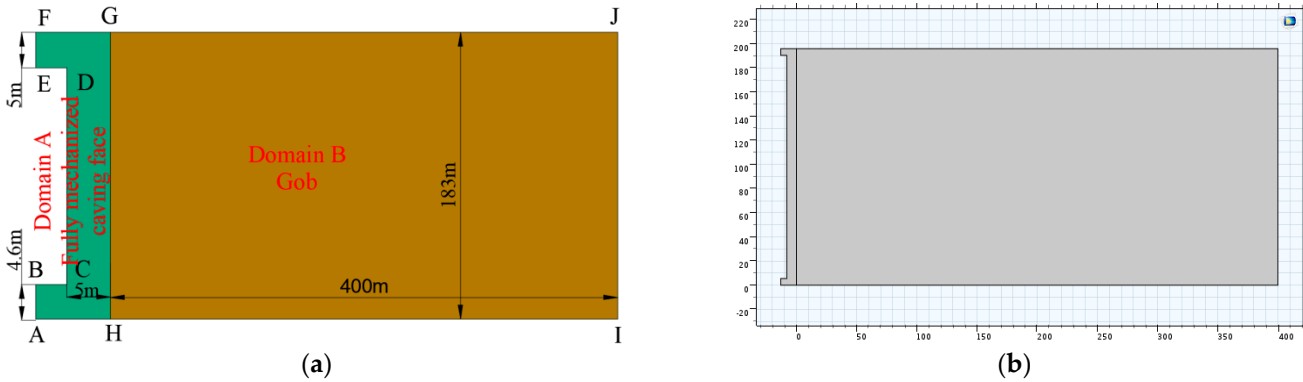

**Figure 1.** Calculation model. (**a**) Geometric model, (**b**) Simulation model.

The average wind speed in the Nanzhuang mine is 2.08 m/s and the air volume is 1300 $m^3$/min. Domains A and B reflect the working face and the goaf so that they are set as the free fluid area and the porous area, respectively. The working face is 183 m long and 5 m wide. Moreover, the air inlet roadway is located below the working face. The 4.6 m wide return air roadway is located above the working face. The air flow flows into the air inlet roadway and out of the air return roadway through the working face and goaf. The oxygen concentration of the working face is 20.9%, and the pressure difference between the inlet and outlet is 98 Pa. Tables 1 and 2 present the main parameters of the model and the imposed boundary conditions in the simulation, respectively.

**Table 1.** Model parameters.

| Description | Parameter | Values and Units |
|---|---|---|
| Density of the gas | $\rho$ | 0.85 kg/$m^3$ |
| Differential pressure of air flow | $\Delta p$ | 98 Pa |
| Dynamic viscosity of air flow | $\mu$ | $1.79 \times 10^{-5}$ Pa·s |
| The tunnel air flow initial temperature | $T_0$ | 28 °C |
| Oxygen diffusion coefficient | $D(O_2)$ | $1.6 \times 10^{-4}$ $m^2$/s |
| Average particle size of coal in the goaf | $\overline{d}$ | 15 mm |
| Oxygen consumption coefficient | $\gamma_0$ | 0.16 mol/($m^3$·h) |
| Initial oxygen concentration | $c_{O_2}^0$ | 9.375 mol/$m^3$ |
| Temperature oxidation index | $a$ | $2.4 \times 10^{-5}$ $K^{-1}$ |
| Average dip angle | $\alpha$ | 8° |
| Fully mechanized top coal caving height | $m_1$ | 2.5 m |
| Fully mechanized coal drawing height | $m_2$ | 3.4 m |
| Direct top thickness | $\sum h$ | 2.9 m |
| Gap between rock mass and roof in the collapse zone | $\Delta h$ | 1.2 m |
| Top coal recovery | $C$ | 83% |
| Residual coal crushing expansion coefficient | $K_{Pc}$ | 1.05 |
| Top control distance of the working face | $L$ | 3.3 m |
| periodic pressure step | $L_0$ | 13 m |
| Diffusion coefficient of air | $D_0$ | $3.5 \times 10^{-5}$ $m^2$/s |

**Table 2.** The imposed boundary conditions and initial values.

| Definite Conditions | | Gas Flow | Convection Diffusion |
|---|---|---|---|
| **Initial condition** | | $\Delta p = 98$ Pa | $c(O_2) = 0$ |
| **Boundary condition** | AB | $\begin{cases} ux = -u_x v_{in} \\ uy = -u_y v_{in} \end{cases}$ | $c(O_2)= 9.375$ mol/m$^3$ |
| | EF | $p = p_1 - Q^2 R$ | Zero flux |
| | DE | Zero flux | Zero flux |
| | BC | Zero flux | Zero flux |
| | AH | Zero flux | Zero flux |
| | FG | Zero flux | Zero flux |
| | CD | Zero flux | Zero flux |
| | GH | $p_W = p_1 - (y \cdot Q^2 R / L)$ | Convection flux |
| | GJ | Zero flux | Zero flux |
| | HI | Zero flux | Zero flux |
| | IJ | Zero flux | Zero flux |

*4.2. Simulation Results and Verification*

In order to verify the accuracy of the mathematical model, a beam tube monitoring test was set up in the gob of the isolated-island working face.

4.2.1. Scheme Design

In the present study, a total of 6 measuring points with a space of 30 m were arranged on the 8824 fully mechanized mining face. The monitoring bundle pipe was protected using a steel pipe with an inner diameter of 5 cm. Among the measuring points, three points are in the return air roadway along the outer wall of the return air roadway, with a space of 200 m, while the other points are in the transportation roadway. Figure 2 shows the layout of the monitoring points.

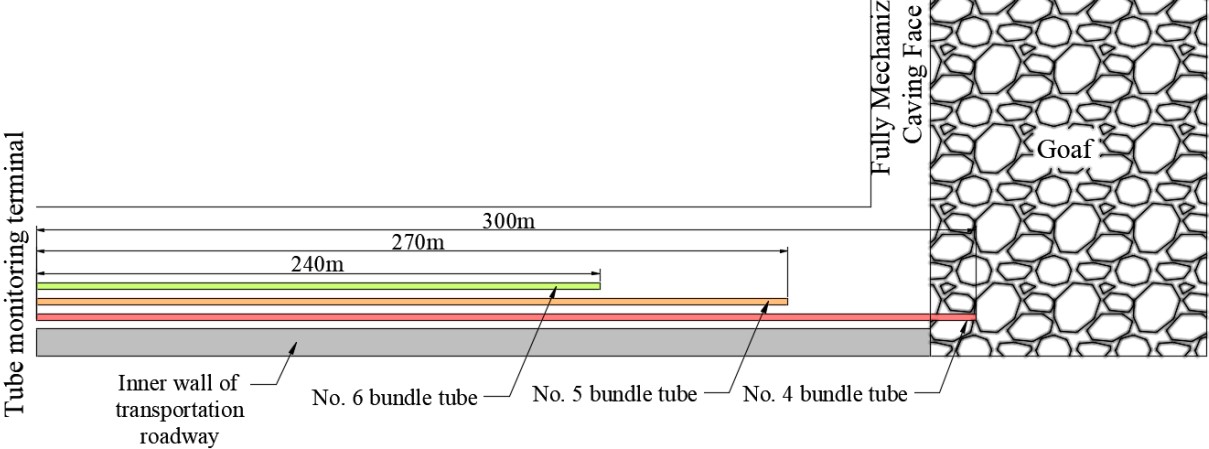

**Figure 2.** Layout of measuring points in the goaf.

The three-beam tubes are bundled together and the extraction sections are aligned. To observe the air extractions, the detection end starts from the steel pipe near the tee in the goaf and ends at the monitoring terminal of the steel pipe. The sampling period is once every two days, and the sampling is performed in the morning. Then the extracted gas samples are analyzed in the afternoon using a gas chromatograph. During the analysis, the measured gas components include the changes of CO and $O_2$.

4.2.2. Result Analysis

Figure 3 illustrates the monitoring results in the bundle pipe at the return air side and inlet air side of the 8824 island working face. When the monitoring point enters 90 m into the goaf, the oxygen concentration at the monitoring point decreases to 18%. As the entry length increases to 193 m, the oxygen concentration changes slightly, and then it decreases rapidly. More specifically, the oxygen concentration decreases to 8.0% for an entry length of 239 m. It is found that the concentration at the return side is lower than that at the inlet side, and the oxygen concentration at the return corner is about 19.56%. When the measuring point enters 86 m into the goaf, the oxygen concentration decreases to 13.66%. When the working face is mined, except for a few areas with an increasing oxygen concentration because of air leakage of the adjacent goaf and roadway, the oxygen concentration decreases continuously. When the entry length of the measuring point reaches 189 m, the oxygen concentration drops to 8.0%.

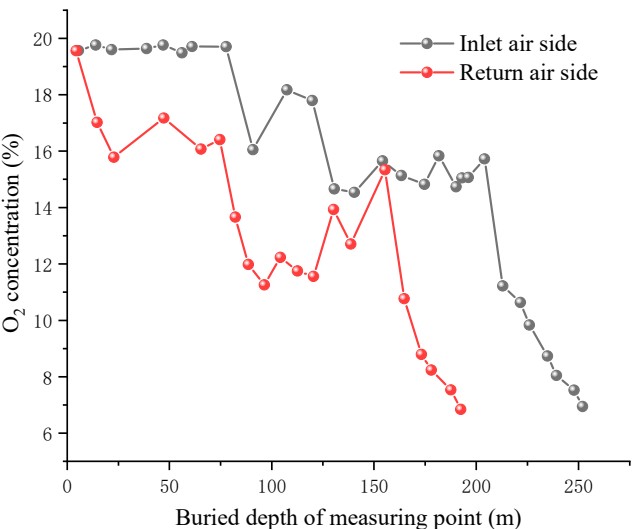

**Figure 3.** Distribution of $O_2$ concentration at measuring points on the return air and the inlet air side.

Figure 4 shows the monitoring results of CO in the bundle pipe at the return air side and inlet air side of the 8824 island working face. It is observed that as the measuring points on the return air side gradually enter the deep part of the goaf, the CO concentration increases first and then decreases slightly. When the measuring point enters 25 m into the goaf, the CO concentration at the measuring point at the return air reaches 64 ppm. Meanwhile, as the entry length of the measuring point increases to about 90–120 m, the CO concentration reaches 144 ppm. Meanwhile, the $O_2$ concentration in this range is about 11.75%, indicating that the air leakage speed here is appropriate, the heat of coal SC compound reaction is accumulated, and the amount of CO is increased, which is in line with the law of coal SC. The heat released from coal SC accumulates, and the CO concentration increases, which is expected from the coal SC. Then the oxygen supply decreases continuously and the reaction weakens. Accordingly, the CO concentration decreases continuously. When the buried depth of the measuring point is longer than 190 m, concentrations of oxygen and CO reach 6.8% and 86 ppm, respectively.

A similar trend with the return side appears for the CO concentration at the inlet side. However, the difference is that the CO concentration at the measuring points on the inlet side is relatively small. With the mining of the working face, no CO was detected within 20 m before the measuring point on the air inlet side. When the entry length reaches 38.8 m, a trace concentration of CO was detected, reflecting slight oxidation of residual coal. When the entry length increases to about 107–164 m, the CO concentration gradually increases. Furthermore, the highest CO concentration (89 ppm) occurs at an entry length of 234 m.

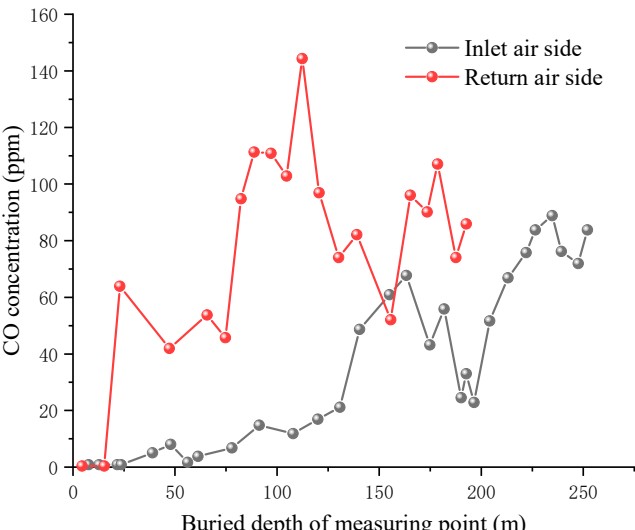

**Figure 4.** Distribution of CO concentration at measuring points on the return air and the inlet air side.

According to the O$_2$ concentration results monitored by the beam tube in the goaf, the three zones of SC in the goaf are divided according to the oxygen concentration of 8.0% and 18.0%. The comparison between the numerical simulation results and the field test results is shown in Table 3.

Table 3 reveals that the measured and calculated heat dissipation zone on the air inlet side is 107 m and 109 m long, respectively, indicating that the calculation error is less than 2%. Moreover, the average calculation error in the suffocation zone and the average error of the maximum width of the oxidation zone are 4.6% and 9.8%, respectively. It is inferred that the performed simulation is consistent with the experiment and the model accuracy is verified. According to the numerical simulation and field test results, the three zones ranges of the goaf are determined. It is observed that the ranges of heat dissipation and oxidation zones in the inlet side are about 0–107 m and 107–239 m, respectively and the rest is the range of the asphyxia zone. Moreover, the ranges of these three zones in the return air side are about 0–13 m, 13–189 m, and beyond 189 m, respectively. The ranges of these tree zones in the middle of the goaf are about 0–52 m, 52–213 m, and beyond 213 m, respectively. Figure 5 shows the three zones in the goaf.

**Table 3.** The comparison of three zones in the goaf of the 8824 isolated-island working face.

| Means | Location | Heat Dissipation Zone (m) | Oxidation Spontaneous Combustion Zone (m) | Suffocation Zone (m) | Maximum Width of the Oxidation Zone (m) |
|---|---|---|---|---|---|
| **Field measurement** | Inlet air side | <107 | 107–239 | >239 | 132 |
| | Return air side | <13 | 13–189 | >189 | 176 |
| **Numerical simulation** | Inlet air side | <109 | 109–228 | >228 | 119 |
| | Middle part | <52 | 52–213 | >213 | 161 |
| | Return air side | <19 | 19–187 | >187 | 168 |

*4.3. Analysis of Influence of Different Air Volume on Three Zones of SC in Goaf*

Figure 6 shows the oxygen concentration contour in a goaf with different wind speeds, including 0.98, 1.48, 2.08, and 3.12 m/s, which correspond to air volumes of 500, 900, 1300, and 1900 m$^3$/min, respectively. The obtained results reveal that the pressure distribution in the goaf is almost symmetrical and gradually decreases from the air inlet side to the air return side. Moreover, the pressure profiles gradually reduce from the working face to the deeper parts. It is observed that the oxygen concentration at the air inlet side is remarkably higher than that at the air return side. Considering the oxygen concentration as an indicator, the goaf can be divided into 18.0% and 8.0% parts [14]. Figure 6 reveals

that as the air supply in the working face gradually increases, the heat dissipation zone, oxidation SC zone, and the suffocation zone move to deeper parts of the goaf. The scope of the oxidation zone, which reflects the scope of the SC risk area, shows an expanding trend with the excessive airflow. On the contrary, when the air volume of the working face decreases, these zones move to the working face and the scope of the oxidation zone decreases gradually.

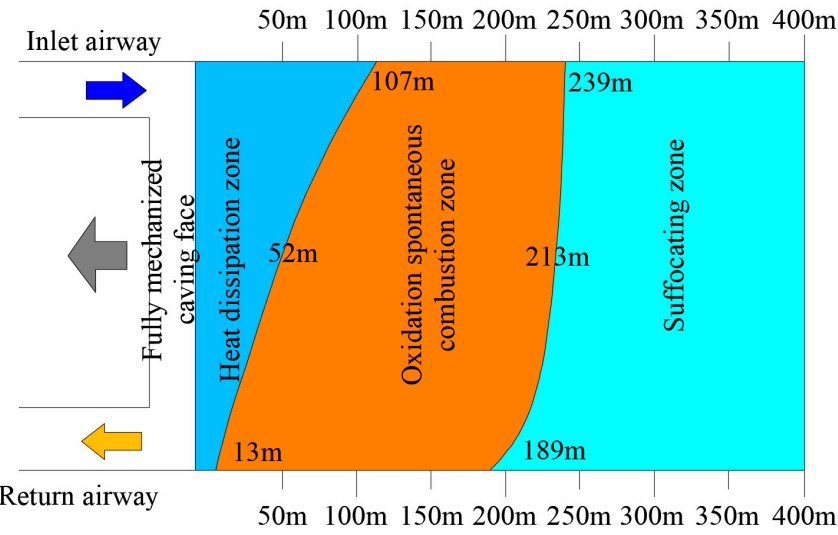

**Figure 5.** Distribution of three zones of SC in the goaf of the isolated-island working face.

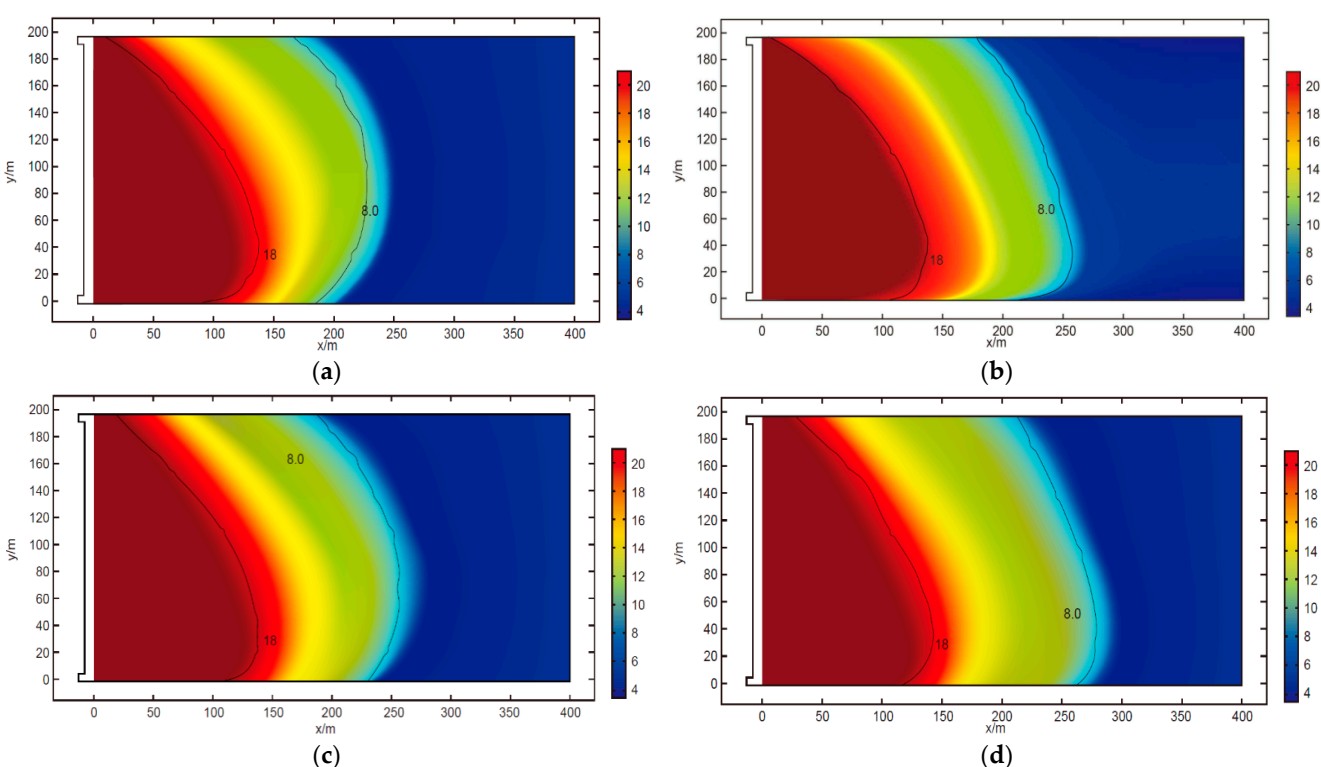

**Figure 6.** Oxygen concentration contours with different air volumes. (**a**) 500 m³/min (equals to a wind speed of 0.98 m/s), (**b**) 900 m³/min (equals to a wind speed of 1.48 m/s), (**c**) 1300 m³/min (equals to a wind speed 2.08 m/s), (**d**) 1900 m³/min (equals to a wind speed 3.12 m/s).

Based on the obtained results, variations of the three zones were analyzed after adjusting the air volume and reaching steady-state conditions. Table 4 shows SC divisions

in goaf for different air volumes, including 500, 900, 1300, and 1900 m³/min, which is equal to the wind speed of 0.98, 1.48, 2.08, and 3.12 m/s, respectively. It is observed that as the air volume increases, the heat dissipation, oxidation, and suffocation zones in the goaf increase in varying degrees. The performed analyses reveal that in the studied cases, the maximum and minimum width of the oxidation zone occurs on the return air side and the inlet air side, respectively.

**Table 4.** Distribution of three zones of goaf during air volume regulation.

| Air Volume (m³/min) | Location | Heat Dissipation Zone (m) | Oxidation Spontaneous Combustion Zone (m) | Suffocation Zone (m) | Maximum Width of the Oxidation Zone (m) |
|---|---|---|---|---|---|
| **500** | A1 | <87 | 87–179 | >179 | 92 |
| | A2 | <31 | 31–165 | >165 | 134 |
| | A3 | <10 | 10–159 | >159 | 149 |
| **900** | A1 | <98 | 98–201 | >201 | 103 |
| | A2 | <42 | 42–189 | >189 | 147 |
| | A3 | <12 | 12–168 | >168 | 156 |
| **1300** | A1 | <109 | 109–228 | >228 | 119 |
| | A2 | <52 | 52–213 | >213 | 161 |
| | A3 | <19 | 19–187 | >187 | 168 |
| **1900** | A1 | <116 | 116–262 | >262 | 146 |
| | A2 | <61 | 61–241 | >241 | 180 |
| | A3 | <27 | 27–210 | >210 | 183 |

Note: A1, A2, and A3 are inlet air side, middle part, and return air side, respectively.

Figure 7 shows the distribution of air volume in the three zones of the goaf. It is observed that as the air volume increases, the corresponding maximum width of the oxidation increases. When the air volume exceeds a certain value, the width of the oxidation zone at the return air side approaches a fixed value, while that at the inlet air side increases continuously. Meanwhile, Figure 7b shows that as the air volume increases, the ranges of the three zones and the heat dissipation zone in the goaf increase gradually. When the air volume exceeds 1600 m³/min, the range of the tropical zone gradually becomes gentle. The range of the oxidation zone has a linear correlation with the air volume. When the air volume is less than 2100 m³/min, the range of the suffocation zone increases in a power function with the air volume.

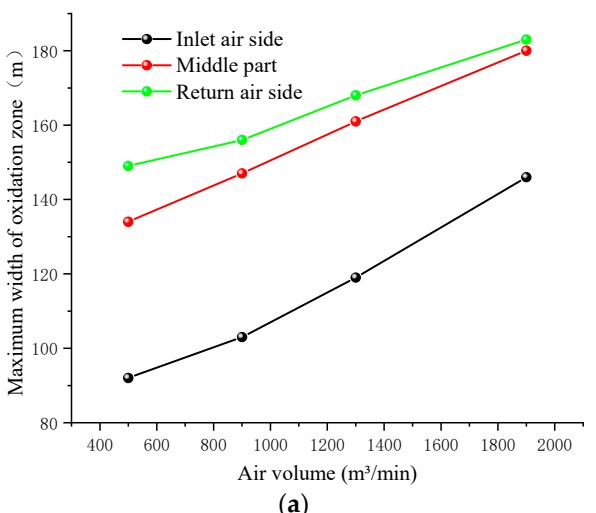

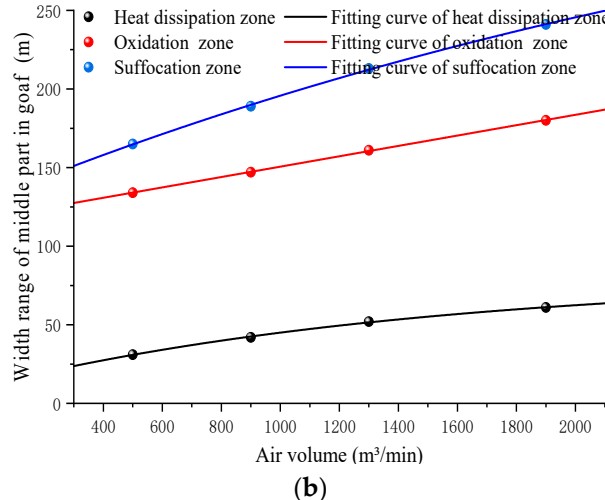

(a)                                                                                              (b)

**Figure 7.** Distribution of the three zones in the goaf against the air volume. (**a**) Distribution of the maximum width of the oxidation zone against the air volume, (**b**) distribution of three zones in the goaf against the air volume.

Under the condition of reasonable ventilation, as the air supply volume of the working face reduces, the oxidation temperature zone moves to the vicinity of the working face, thereby reducing the oxidation SC zone scope. Meanwhile, the dangerous area of the natural ignition of the residual coal moves to the scope of the suffocation zone, and the SC of the residual coal is restrained independent of other fire prevention and extinguishing methods.

## 5. Prediction of SC Index Gas in Goaf

### 5.1. Experimental Conditions

To detect an SC index gas in the goaf of the 8824 working face, low-temperature oxidation and SC of the coal seam number 15 were simulated. The test samples were prepared 50 m away from the belt roadway. The external oxidation part of large samples was removed, cored, and crushed. Consequently, particle sizes of around 0–1 mm, 1–3 mm, 3–5 mm, and 5–10 mm were prepared for the test. The mixing ratio of the four particle sizes is 1:1:1:1. During the test, dry air with a flow rate of 130 mL/min entered the tank for a period of time to increase the temperature at a rate of 0.5 °C/min. When the desired temperature was achieved, the gas sampling was carried out and the gas composition was analyzed after cooling for 2 min.

### 5.2. Experimental Equipment

Figure 8 shows the configuration of the test equipment. The test system mainly consisted of a gas supply, a temperature-programmed system, and a gas analysis system. The gas supply part included an air pump, a flowmeter, a 5 m preheating copper pipe, and high-temperature resistant gas pipes. Moreover, the programming system included a temperature-programmed furnace, a coal sample tank, and two thermometers. Finally, the gas analysis system included a gas chromatograph, a drying tube, and needle tubes.

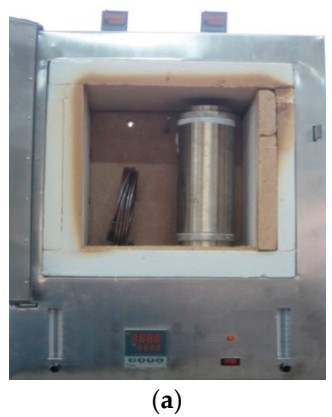
(**a**)

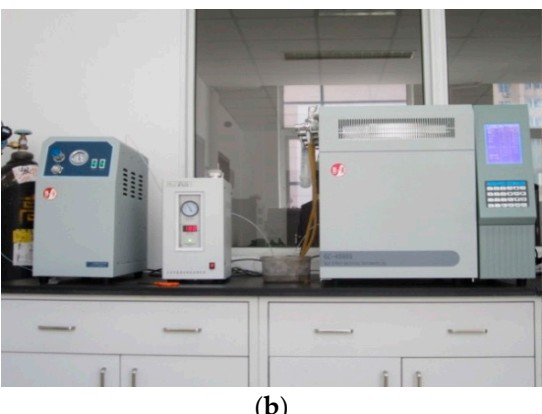
(**b**)

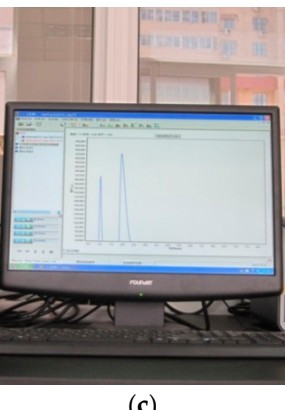
(**c**)

**Figure 8.** Test setup of the coal SC marker gas. (**a**) Temperature-programmed furnace, (**b**) gas chromatograph, (**c**) analysis software.

### 5.3. Experimental Results

In this section, we present the results of the study of the SC index gas of the coal seam number 15 and the optimization of the early warning index gas of SC to ensure safe and efficient coal mining. In this regard, Table 5 presents the gas concentrations originating from coal oxidation and heating.

**Table 5.** Gas concentrations during coal sample heating and oxidation.

| Serial Number | Temperature (°C) | $O_2$ (%) | CO (ppm) | $CH_4$ (ppm) | $CO_2$ (ppm) | $C_2H_4$ (ppm) | $C_2H_6$ (ppm) | $C_3H_8$ (ppm) |
|---|---|---|---|---|---|---|---|---|
| 1 | 20 | 20.48 | 64.64 | 1131 | 767 | 0.00 | 0.00 | 0.00 |
| 2 | 30 | 20.07 | 138 | 8563 | 751 | 0.00 | 0.00 | 0.00 |
| 3 | 40 | 19.85 | 216 | 11,130 | 3492 | 0.00 | 0.00 | 0.00 |
| 4 | 50 | 18.82 | 344 | 12,840 | 11,610 | 0.00 | 0.00 | 0.00 |
| 5 | 60 | 17.74 | 653 | 15,290 | 12,960 | 0.00 | 0.00 | 0.00 |
| 6 | 70 | 17.05 | 792 | 16,100 | 23,570 | 0.00 | 0.00 | 0.00 |
| 7 | 80 | 16.84 | 985 | 17,100 | 54,350 | 0.00 | 0.00 | 0.00 |
| 8 | 90 | 16.52 | 1954 | 19,270 | 71,300 | 0.00 | 0.00 | 0.00 |
| 9 | 100 | 16.18 | 4179 | 28,330 | 79,620 | 0.00 | 15.15 | 12.67 |
| 10 | 110 | 15.44 | 6505 | 30,400 | 75,600 | 10.04 | 50.56 | 28.92 |
| 11 | 120 | 12.04 | 7895 | 39,500 | 78,140 | 39.75 | 107.40 | 39.04 |
| 12 | 130 | 9.820 | 8954 | 45,470 | 80,900 | 59.48 | 122.79 | 46.15 |
| 13 | 140 | 5.299 | 11,440 | 64,270 | 86,040 | 109.12 | 148.30 | 57.40 |
| 14 | 150 | 4.689 | 12,380 | 77,650 | 87,580 | 145.93 | 229.90 | 65.46 |
| 15 | 160 | 3.172 | 15,300 | 92,320 | 82,100 | 194.04 | 260.17 | 75.47 |
| 16 | 170 | 1.416 | 18,950 | 109,500 | 85,300 | 232.95 | 297.41 | 149.83 |
| 17 | 180 | 0.714 | 26,400 | 213,800 | 95,270 | 286.68 | 336.08 | 226.39 |
| 18 | 200 | 0.527 | 30,700 | 356,100 | 101,400 | 369.14 | 412.00 | 310.33 |

### 5.3.1. Generation of Index Gases

Generally, coal gas contains $CH_4$ and it cannot be used as the natural ignition index gas of the coal seam number 15. Figure 9 shows the concentration of different index gases against the coal temperature. It is observed that CO gas is produced only when the coal temperature exceeds 20 °C. Figure 9 reveals that when the coal temperature exceeds 90 °C, the production rate increases rapidly, and the oxidation reaction occurs. Since the generation of CO can better reflect the SC characteristics of Nanzhuang mine coal seam number 15, it can be used as an effective index in all tests. Meanwhile, it is found that when the coal temperature is 20 °C, the $CO_2$ concentration is 655 ppm, indicating that $CO_2$ gas has confined in coal pores. Since the gas production does not have an exponential correlation with the coal temperature, it cannot be used as the SC index gas.

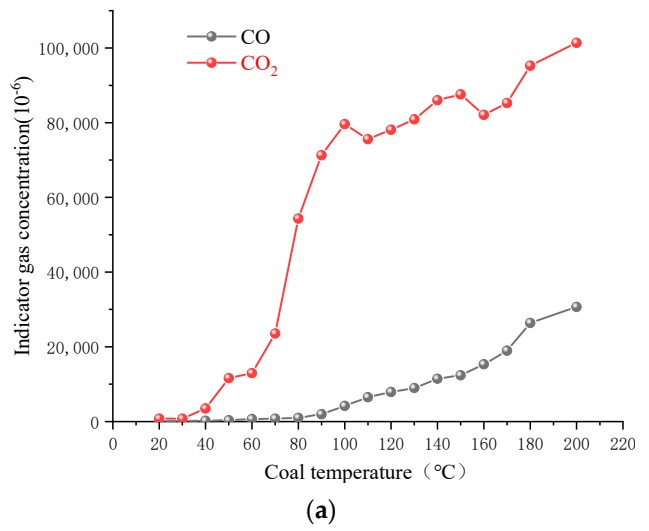

(**a**)

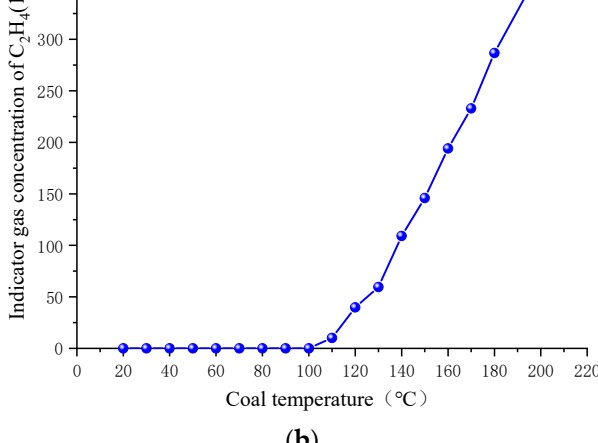

(**b**)

**Figure 9.** *Cont.*

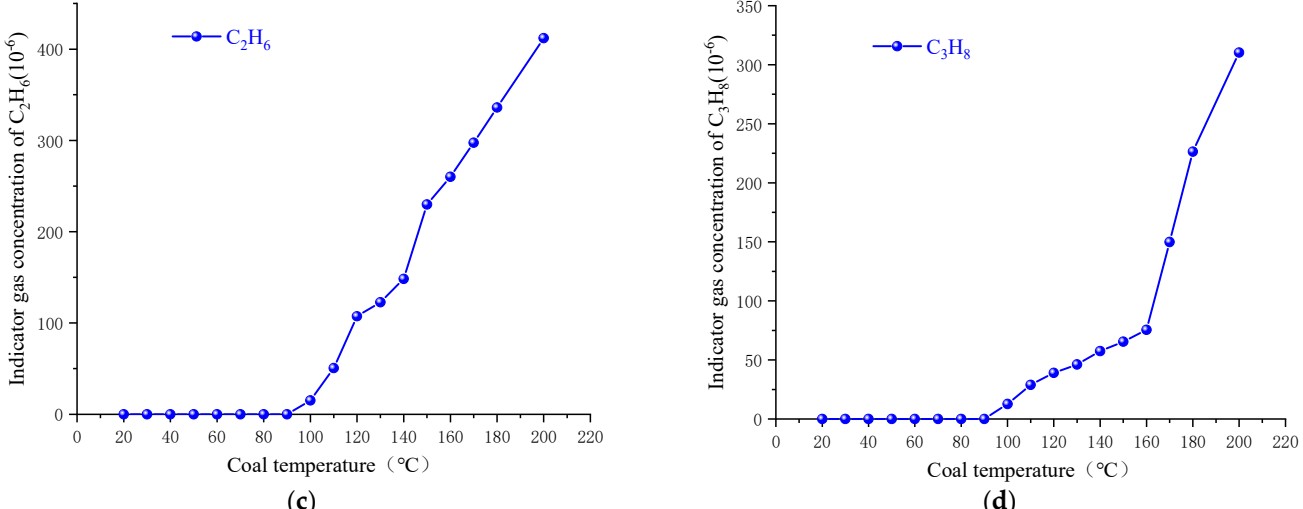

**Figure 9.** Concentration distribution of different index gases in coal seam against the coal temperature. (**a**) CO and $CO_2$, (**b**) $C_2H_4$, (**c**) $C_2H_6$, (**d**) $C_3H_8$.

Figure 9b–d show the concentration of $C_2H_4$, $C_2H_6$, and $C_3H_8$ against the coal temperature. It is observed that when the initial temperature of $C_2H_4$ is 110 °C and the concentration is 10 ppm, there is an exponential correlation between the production rate of $C_2H_4$ and the coal temperature. For example, when the coal temperature reaches 200 °C, the corresponding $C_2H_4$ concentration reaches 369 ppm. The appearance of $C_2H_4$ indicates that the coal SC has entered the accelerated oxidation stage. Accordingly, it is concluded that $C_2H_4$ can be used as an index gas in the coal seam number 15. Furthermore, Figure 9c indicates that the temperature of $C_2H_6$ is 100 °C, its concentration is 15ppm, and its production has a direct correlation with the coal temperature. It is inferred that $C_2H_6$ gas can be used as an auxiliary SC index for coal seam number 15. Figure 9d shows that when the temperature and concentration of $C_3H_8$ are 100 °C and 12 ppm, respectively, its production rate has a direct correlation with the coal temperature. For example, the concentration of $C_3H_8$ at 200 °C is 310 ppm and the production rate has a direct correlation with coal temperature. Therefore, $C_3H_8$ can also be used as an auxiliary index to monitor SC in coal seam number 15. $C_2H_2$ is also the product of coal entering the violent oxidation stage, but $C_2H_2$ did not appear before the coal temperature of 200 °C in this experiment. However, if $C_2H_2$ is detected underground, it indicates that the coal temperature has exceeded 200 °C. Therefore, $C_2H_2$ can also be used as one of the indicator gases in the stage of severe oxidation of coal seams.

5.3.2. Gas Ratio Analysis

The performed analyses demonstrate that CO is a sensitive marker gas in the coal SC. Due to the complex underground conditions, the place where CO is detected is not necessarily a high-temperature point. CO is diluted by a large amount of seepage air, and its concentration depends on the wind flow. Therefore, it is a challenge to judge the actual SC of loose coal originating from the CO concentration. Figure 10a shows the distribution of the $CO/CO_2$ ratio of Nanzhuang mine coal seam number 15 against the coal temperature, indicating that there is no general rule. It is concluded that the ratio of $CO/CO_2$ cannot be used as an appropriate ignition index in coal seam number 15.

Alkene to alkane ratio refers to the ratio of alkene gas concentration to alkane concentration of a certain carbon chain greater than or equal to the alkene in fire gas. In this section, $C_2H_4/C_3H_8$ and $C_2H_4/C_2H_6$ ratios are analyzed. Figure 10b shows that $C_2H_4/C_3H_8$ ratio has poor regularity and the same value may correspond to multiple temperatures. Therefore, $C_2H_4/C_3H_8$ cannot be used as an auxiliary index in coal seam number 15 while there is a direct correlation between $C_2H_4/C_2H_6$ ratio and the coal temperature.

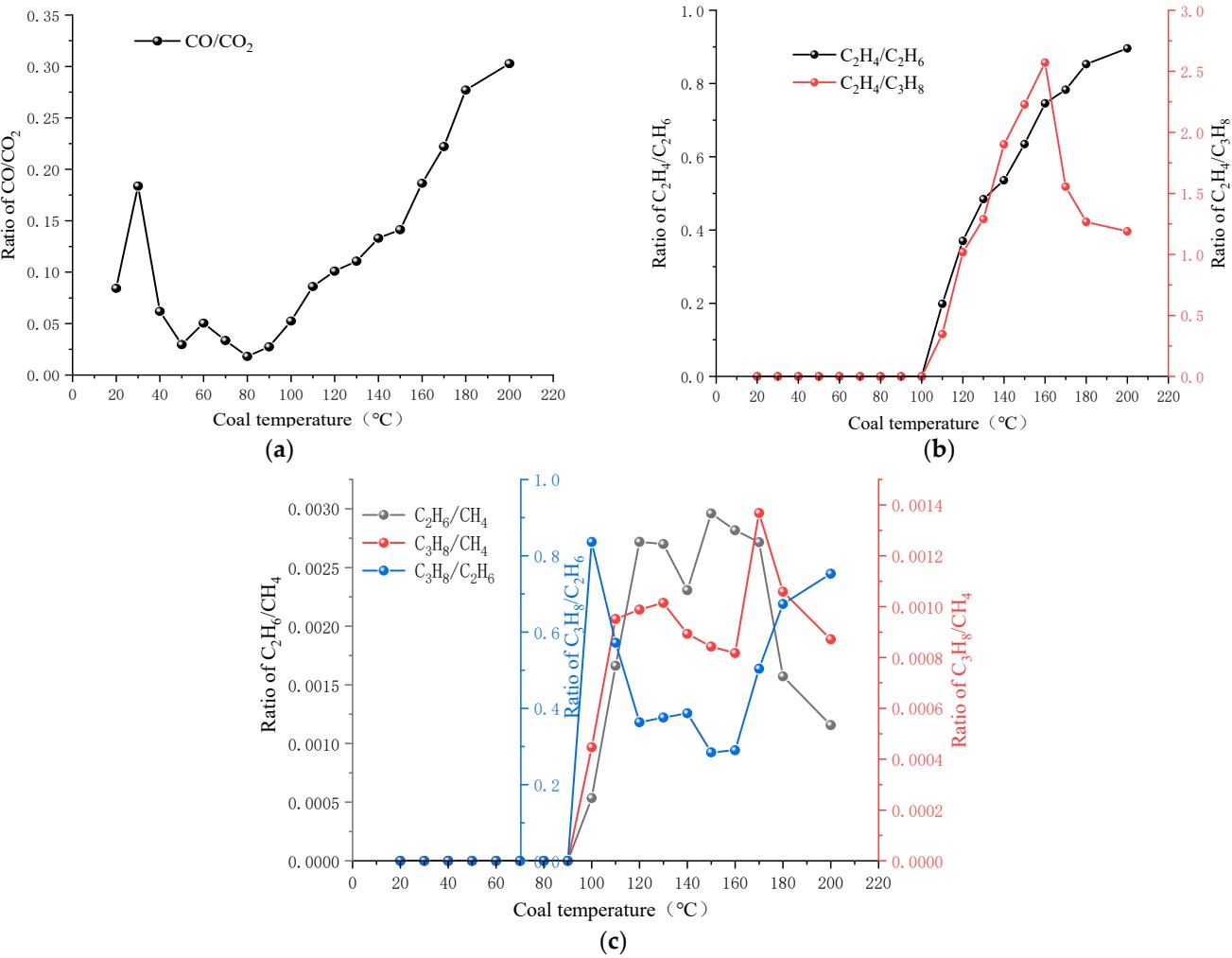

**Figure 10.** Distribution of different gas ratios against the coal seam. (**a**) $CO/CO_2$ ratio, (**b**) alkene ratio, (**c**) alkane ratio.

Alkane ratio refers to the ratio of the concentration of a single component of a long-chain alkane to the concentration of methane or ethane in the alkane gas within the approximate range of $C_1–C_4$ in the fire gas. In this regard, $C_2H_6/CH_4$, $C_3H_8/CH_4$, and $C_3H_8/C_2H_6$ ratios are analyzed and the results are presented in Figure 10c. It is observed that $C_2H_6/CH_4$, $C_3H_8/C_2H_6$, and $C_3H_8/CH_4$ ratios have poor regularity, and the same ratio corresponds to different temperatures. Therefore, $C_2H_6/CH_4$, $C_3H_8/C_2H_6$, and $C_3H_8/CH_4$ ratios cannot be used as an indicator in coal seam number 15.

5.3.3. Optimization Analysis of Coal Seam SC Index in the Gas System

In this section, we present the optimization experiments using the coal SC marker gas index. The gas generation and coal SC characteristic parameters are analyzed in the coal temperature range of around 20–200 °C and the obtained results are presented in Table 6.

**Table 6.** Optimization results of coal seam SC index gas.

| Index Classification | Indicator Name | Initial Temperature (°C) | Concentration (Ratio) (ppm) | | |
|---|---|---|---|---|---|
| | | | 20~60 °C | 60~100 °C | 100~200 °C |
| **Main index** | CO | 20 | 64.64~653 | 653~4179 | 4179~30,700 |
| | $C_2H_4$ | 110 | 0 | 0~10.04 | 10.04~369.14 |
| **Auxiliary index** | $C_2H_6$, $C_3H_8$, $C_2H_4/C_2H_6$ | | | | |

In coal seam number 15 of Nanzhuang mine, CO and $C_2H_4$ are considered as index gases, supplemented by $C_2H_6$, $C_3H_8$, and $C_2H_4/C_2H_6$ to master the coal SC. The initial coal temperature of CO gas is 20 °C, indicating that a low-temperature oxidation reaction has occurred in coal seam number 15. The initial concentrations of $C_2H_4$, $C_2H_6$, and $C_3H_8$ are small and vulnerable to air flow. Detection of these compositions shows that the coal temperature has exceeded 100 °C and the coal has entered the stage of accelerated oxidation. $C_2H_4/C_2H_6$ ratio has a direct correlation with coal temperature so it can be used as an auxiliary index of natural ignition in the coal seam.

## 6. Conclusions

To resolve the SC problem in the goaf of the isolated-island working face, a multi-physical field coupling model was established and different affecting parameters, including the air inlet velocity and concentration of index gases, were analyzed experimentally. Based on the performed analyses, the main achievements of the present study can be summarized as follows:

1. Based on the oxygen concentration, the goaf space can be divided into three zones, including the loose zone, oxidation temperature rise zone, and asphyxia zone. It was found that as the air supply increases, the three zones move to deep parts of the goaf, and the range of the SC risk area increases. When the air volume of the working face decreases, the three zones move to the vicinity of the working face and the range of oxidation temperature rise zone decreases gradually, which provides technical support to restraining SC in the goaf and adjust the air inlet velocity.

2. Taking 8824 working face as the research object, the three SC zones were studied experimentally and numerically. It was found that the ranges of the three zones on the air inlet side are around 0–107 m, 107–239 m, and beyond 239 m, respectively. Moreover, the ranges of the zones on the return air side are around 0–13 m, 13–189 m, and beyond 189 m, respectively. In the middle of the goaf, the ranges of the three zones are around 0–52 m, 52–213 m, and beyond 213 m, respectively. The present study provides theoretical and technical support to prevent and control SC in the goaf.

3. Through the temperature-programmed experiment, the index gases of the coal seam number 15 were determined, and the coal SC was mastered by $C_2H_6$, $C_3H_8$, and $C_2H_4/C_2H_6$. The initial coal temperature of CO gas is 20 °C, indicating that a low-temperature oxidation reaction has occurred in the coal seam number 15 of the Nanzhuang mine. On the other hand, detection of $C_2H_4$, $C_2H_6$, and $C_3H_8$ compositions reveals that the coal temperature has exceeded 100 °C and the coal has entered the accelerated oxidation stage. It provides a theoretical basis for the prediction system of goaf spontaneous combustion in an isolated-island working face.

**Funding:** This research was funded by the National Nature Fund project of China (No. 51804161), and National Nature Fund project of China (No. 52174188).

**Acknowledgments:** This work was supported by the National Nature Fund project of China: Study on the Flame Retardant Mechanism of Inert Gas Dynamic Replacement of Adsorbed Oxygen in Coal at Atmospheric Pressure (51804161), and National Nature Fund project of China: Study on the Failure Mechanism of Rock Instability on High and Steep Slope of Water Storage Open-pit Mine under Seepage-stress Coupling (52174188), which are gratefully acknowledged. Meanwhile, we sincerely appreciate that editors and reviewers have spent their precious time reviewing this paper and giving critical and valuable comments.

**Conflicts of Interest:** The author declares that there are no conflict of interest regarding the publication of this paper. And all relevant data are within the manuscript.

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
