# Peer review of "Investigation of Spontaneous Combustion Zones and Index Gas Prediction System in Goaf of “Isolated Island” Working Face"

_fire, doi:10.3390/fire5030067_

Round 1

Reviewer 1 Report

I have read the whole article and I think it is a very interesting idea to investigate the spontaneous combustion zones and index gas prediction system in goaf of “isolated island” working face. The author had given the range of heat dissipation zone, oxidation zone, and the asphyxis zone on both side of the working face with a multi-pfysics model coupled with gas flow field and gas concentration field. A coal programmed temperature rise experiment was carried out to improve the prediction idex gas system of SC. There are several question which the autor should make clear.

(a) What is the “Mathematical model” part used for? Does the part refect the “isolated island” working face?

(b) The differce between the “isolated island” working face and regular working face should be given.

Reviewer 2 Report

The paper presents the interesting investigations. The contents and results are comprehensive. But it is need revision according comments:

  1. The Authors should add explanations about modern approaches in the investigated field.
  2. Please, explain the choice of research methods and the main suggestions.
  3. Introduction section can be improved using more strong structure, subsections and more detailed analysis of previous research results.
  4. The choice of the research object and input parameters must be justified.
  5. The type of research objects and real technologies can differ significantly. Additional comments are needed in the article.
  6. It is necessary to perform mathematical processing of research results in a dimensionless form for their generalization.
  7. Findings and recommendations need to be segregated and clearly structured.
  8. It is advisable in the article to present the results of a comparison of data from other authors, since there are many results of similar studies.

Correction of the manuscript will enable to improve the quality of the submitted material.

Round 2

Reviewer 1 Report

This manuscript has been modified for publication.

Reviewer 2 Report

Article can be accepted

This manuscript is a resubmission of an earlier submission. The following is a list of the peer review reports and author responses from that submission.